# Rutin/Sulfobutylether-β-Cyclodextrin as a Promising Therapeutic Formulation for Ocular Infection

**DOI:** 10.3390/pharmaceutics16020233

**Published:** 2024-02-05

**Authors:** Federica De Gaetano, Martina Pastorello, Venerando Pistarà, Antonio Rescifina, Fatima Margani, Vincenzina Barbera, Cinzia Anna Ventura, Andreana Marino

**Affiliations:** 1Department of Chemical, Biological, Pharmaceutical and Environmental Sciences, University of Messina, Viale Ferdinando Stagno d’Alcontres 31, 98166 Messina, Italy; fedegaetano@unime.it (F.D.G.); martina.pastorello@studenti.unime.it (M.P.); 2Department of Pharmaceutical and Health Sciences, University of Catania, Viale Andrea Doria 6, 95125 Catania, Italy; vpistara@unict.it (V.P.); arescifina@unict.it (A.R.); 3Department of Chemistry, Materials and Chemical Engineering “G. Natta”, Politecnico di Milano, Via Mancinelli 7, 20131 Milano, Italy; fatima.margani@polimi.it (F.M.); vincenzina.barbera@polimi.it (V.B.)

**Keywords:** rutin, sulfobutylether-β-cyclodextrin, antibacterial activity, resistant strains, biofilm

## Abstract

Ocular pathologies present significant challenges to achieving effective therapeutic results due to various anatomical and physiological barriers. Natural products such as flavonoids, alone or in association with allopathic drugs, present many therapeutic actions including anticancer, anti-inflammatory, and antibacterial action. However, their clinical employment is challenging for scientists due to their low water solubility. In this study, we designed a liquid formulation based on rutin/sulfobutylether-β-cyclodextrin (RTN/SBE-β-CD) inclusion complex for treating ocular infections. The correct stoichiometry and the accurate binding constant were determined by employing SupraFit software (2.5.120) in the UV-vis titration experiment. A deep physical–chemical characterization of the RTN/SBE-β-CD inclusion complex was also performed; it confirmed the predominant formation of a stable complex (K_c_, 9660 M^−1^) in a 1:1 molar ratio, with high water solubility that was 20 times (2.5 mg/mL) higher than the free molecule (0.125 mg/mL), permitting the dissolution of the solid complex within 30 min. NMR studies revealed the involvement of the bicyclic flavonoid moiety in the complexation, which was also confirmed by molecular modeling studies. In vitro, the antibacterial and antibiofilm activity of the formulation was assayed against *Staphylococcus aureus* and *Pseudomonas aeruginosa* strains. The results demonstrated a significant activity of the formulation than that of the free molecules.

## 1. Introduction

Ocular infections are a worldwide health problem. If not treated properly, they can become worse, damaging the anatomic structure of the eye and leading to permanent vision loss or blindness [1].

Common ocular infections caused by microorganisms include conjunctivitis, keratitis, cellulitis, endophthalmitis, and dacryocystitis. Among these, keratitis is a devastating disease that is responsible for up to 2 million annual cases of blindness globally [2]. The main microorganisms causing ocular infections are bacteria [3]. Among these, *S. aureus* and *P. aeruginosa* are the most common pathogens [4]. *S. aureus* is a Gram-positive bacterium that is a leading cause of conjunctivitis, keratitis, and endophthalmitis [5]. This species is included in the ESKAPE group (*Enterococcus faecium*, *Staphylococcus aureus*, *Klebsiella pneumoniae*, *Acinetobacter baumannii*, *Pseudomonas aeruginosa*, and *Enterobacter* species). These bacteria are involved in infections and are categorized by multidrug resistance. Incidence of ocular infection caused by methicillin-resistant *S. aureus* (MRSA) has increased considerably over the last two decades [6] and this represents a significant global health concern mainly in hospital (HA-MRSA) and community settings (CA-MRSA) [5,6]. The Surveillance Network in the United States reported an increase in the percentage of MRSA among *S. aureus* ocular infections, from 29.5% in 2000 to 41.6% in 2005. In India, the incidence of MRSA increased from 9% in 2007 to 38% in 2017 [7]. Moreover, *S. aureus* can form biofilm which aggravates the problem of antimicrobial resistance. Biofilm is a structured consortium of bacteria embedded in a self-produced polymer matrix of polysaccharides, proteins, and DNA; it represents a physical barrier against drugs and host immune responses. *P. aeruginosa* is a Gram-negative bacterium included in the ESKAPE group, which is resistant to treatment with most antibiotics and causes vision-threatening ocular infections. Moreover, due to its predisposition to form biofilms, *P. aeruginosa* is the major cause of keratitis in contact-lens wearers [8]. In the United States, *P. aeruginosa* incidence may be as high as 70% for contact lens-associated keratitis. Treatment for several ocular surface diseases comprises eye drops containing antibiotics. However, this can stimulate changes in the healthy eye microbiota and contribute to increasing the resistance of pathogenic strains [9]. Currently, treatment for ocular surface infections involves the use of topical antibiotics such as fluoroquinolones, tetracycline, or chloramphenicol. Alternatively, fortified antibiotics (e.g., cefazolin with gentamicin or tobramycin) may be used in cases of severe infection [5,10,11].

As a response to rising antibiotic resistance, new therapeutic strategies are in development. Over the years, interest in plant extracts and their metabolites has risen within the scientific community [12,13]. Rutin (RTN) (Figure 1), which is a flavonoid isolated from plants or fruits, showed antibacterial activity against *S. aureus*, *A. baumannii*, and *P. aeruginosa* [14]. Some authors have suggested that RTN interferes with the membranes of *P. aeruginosa* and MRSA cells, causing the release of proteins and nucleic acids. Moreover, it has been demonstrated to decrease biofilm biomass of *P. aeruginosa* and MRSA catheter biofilms with a mechanism leading to the reduction of cell viability, exopolysaccharide, and extracellular DNA levels [15]. Furthermore, it has been demonstrated that RTN potentiates the antibacterial activity of different antibiotics against MRSA strains [4].

Despite these exciting attributes, unfavorable physical–chemical properties of RTN—particularly water solubility [16,17]—prevent its employment in therapy. It is highly insoluble in water and biological fluids in the doses in which it is to be administered, showing poor and erratic bioavailability [18]. Cyclodextrins (CDs) could be used to overcome this drawback. They are cyclic oligosaccharides that are able to complex apolar drugs and effectively improve their physicochemical properties, including water solubility [19,20,21,22,23,24,25,26] and biological effects [27,28,29]. 

A search of the literature shows that many studies have been performed concerning the preparation and physical–chemical characterization of the inclusion complexes between RTN and native or modified CDs [30,31]. In some cases, in vitro or in vivo biological activities were reported. For example, the complexation of RTN within native β-CD [32] and hydroxypropyl-β-CD (HP-β-CD) [33,34] improves its antioxidant and antimicrobial effects in vitro [34,35]. Furthermore, RTN/β-CD and RTN/dimethyl-β-CD showed improved antioxidant activity compared with free RTN in the DPPH free radical scavenging assay [36]. Recently, RTN/β-CD and RTN/HP-β-CD inclusion complexes have demonstrated good antiproliferative activity in different cultured cancer cells [37,38]. Oral administration of RTN/HP-β-CD inclusion complex produced increased bioavailability of RTN from the gastrointestinal tracts of beagle dogs due to increased water solubility, faster dissolution rate, and gastrointestinal stability of the drug [39].

To the best of our knowledge, although substantial literature concerning the complexation of RTN with different modified CDs is available, very few papers describe the complexation of RTN with sulfobutyl-ether-β-CD (SBE-β-CD). Furthermore, none of these concern its antimicrobial and antibiofilm activity either in vitro or in vivo. Wu et al. [40] conducted a luminescence study to determine the association constant of RTN/SBE-β-CD inclusion complex and applied this technique to quantify RTN on urine samples. Gozcu et al. made a gel containing RTN/SBE-β-CD inclusion complex as a promising topical system [41], and Zhou et al. [42] reported a UV-vis spectroscopic characterization of the complex. However, neither were biological studies.

SBE-β-CD is an anionic modified CD (Figure 1) with higher water solubility and complexing ability than native β-CD or other modified β-CDs that are usually present in marketed pharmaceutical products. It shows high biocompatibility and non-toxicity, and is approved by the FDA for intramuscular (IM) and intravenous (IV) administration. Fifteen products are on the market today containing SBE-β-CD. This CD shows excellent complexing ability toward different drugs, significantly improving their solubility and dissolution rate and potentiating their biological effects [43]. The complexation of carbamazepine into SBE-β-CD enhances the antiepileptic activity in vivo compared with carbamazepine suspension, due to significant improvement of the drug solubility in gastric fluids, ameliorating absorption and oral bioavailability [44]. Curcumin/SBE-β-CD inclusion complex demonstrated superior antimicrobial activity against *Escherichia coli* and *Staphylococcus aureus* than free curcumin, showing good potential as a treatment option for urinary tract infections [45].

Based on these excellent properties, a more remarkable improvement of physical–chemical and biological properties of RTN could be expected using SBE-β-CD rather than other CDs. Thus, in this work, we developed a liquid formulation based on RTN/SBE-β-CD inclusion complex, which is able to improve the antibacterial and antibiofilm activity of RTN. The formulation is designed to be administered ophthalmically. The inclusion complex was prepared by freeze-drying and characterized by phase-solubility studies, UV-vis, and NMR spectroscopy. Thermogravimetric analysis and X-ray diffraction were performed on the inclusion complex in a solid state. Molecular modeling studies on the RTN/SBE-β-CD inclusion complex were conducted to evaluate the energetic and structural rationalization of the recognition process. Finally, the antibacterial properties of RTN/SBE-β-CD inclusion complex were assayed in comparison with those of free components, and complexed with SBE-β-CD by in vitro studies against *Staphylococcus aureus* and *Pseudomonas aeruginosa*.

## 2. Materials and Methods

### 2.1. Materials

Rutin (RTN, C_27_H_30_O_16_, molecular weight, 610.5 g/mol) and levofloxacin hydrochloride (LVF, C_18_H_20_FN_3_O_4_, molecular weight (MW) 361.37 g/mol) were purchased from Sigma-Aldrich (St. Louis, MO, USA). Sulfobutyl-ether-β-cyclodextrin (SBE-β-CD, CAPTISOL^®^, average degree of substitution seven, average molecular weight, 2162 g/mol) was supplied by CyDex Pharmaceutical (Lenexa, KS, USA). Culture media were purchased from ThermoFisher (Oxoid, Milan, Italy). The other reagents were purchased from Sigma-Aldrich (Milan, Italy) unless otherwise specified in the text.

### 2.2. Preparation of RTN/SBE-β-CD Inclusion Complex and Physical Mixture

RTN/SBE-β-CD inclusion complex was prepared by lyophilization. Initially, 10 mg of RTN (1.64 M^−3^) was weighed and solubilized in 2 mL of MeOH. Subsequently, 71 mg of SBE-β-CD (3.3 M^−3^) was solubilized in 8 mL of H_2_O and added to the methanol solution of RTN. The mixture was magnetically stirred at room temperature for 30 min. The obtained solution was freeze-dried for 72 h (VirtTis Benchtop K Instrument, SP Scientific, Gardiner, NY, USA). 

RTN/SBE-β-CD physical mixture in a 1:2 molar ratio was obtained by carefully mixing an accurately weighed amount of drug and SBE-β-CD in a mortar, until the mixture was homogeneously colored.

### 2.3. UV-Vis Titration

Free RTN (0.09 mg, 0.03 M^−3^) and increasing concentrations of SBE-β-CD (0.03, 0.09, 0.15, 0.3, 0.9, 1.5, 2.1, 3 × 10^−3^ M) were solubilized in a water/methanol mixture (80:20, *v*/*v*) and stirred in the dark at 500 rpm for 24 h, before the analysis. The solutions were analyzed by UV-vis spectroscopy in the spectral range 200–600 nm using a diode array spectrophotometer, StellarNet BLACK-Comet, Model C (Tampa, FL, USA).

### 2.4. Phase-Solubility Studies

Phase-solubility studies of the RTN/SBE-β-CD system were performed by the Higuchi and Connors method [46]. RTN was added to aqueous solutions containing increasing concentrations of SBE-β-CD (0–12 mM) at concentrations exceeding its intrinsic solubility. The obtained suspensions were left to stir in the dark in a thermostatic bath at 25.0 ± 0.5 °C (Telesystem 15.40, Thermo Scientific, Waltham, MA, USA). The bath was equipped with a temperature control unit (Telemodul 40C, Thermo Scientific, USA). After that, the suspensions were filtered through Sartorius Minisart-SRP 15-PTFE filters (0.22 µm, Bedford, MA, USA). The solutions were then diluted with methanol (all final solutions were water/methanol, 80/20, *v*/*v*) and analyzed spectrophotometrically to quantify RTN in solution. A calibration curve prepared in water/methanol solution (20/80, *v*/*v*) at a λmax of 256 nm, with concentrations ranging from 0.00090 to 0.036 mg/mL, was used and an R^2^ value of 0.9953 was obtained.

The experiments were carried out in triplicate to obtain reliable results, and phase solubility diagrams were constructed. The concentration of SBE-β-CD is represented on the abscissas of these diagrams, while the concentration of RTN in solution is represented on the ordinates. Using the following Higuchi and Connors equation (Equation (1)) [46], the apparent 1:1 association constant (K_c_) of the complex was calculated, where S_0_ is the intrinsic water solubility of RTN:(1)$$K_{c}=\frac{Slope}{\left(1-Slope\right)S_{0}},$$

### 2.5. Nuclear Magnetic Resonance Experiments

Samples containing equivalent concentrations (9 mM) of RTN, SBE-β-CD, and RTN/SBE-β-CD inclusion complex were added to a D_2_O/CD_3_OD (7/3, *v*/*v*) solution and transferred to 5 mm NMR tubes. A Varian Unity Inova 500 MHz (11.75 T) instrument was used for all spectra recorded at 300 K. The residual water peak (4.79 ppm) was used as the internal reference to avoid adding external ones that could interact with SBE-β-CD.

### 2.6. Molecular Modeling Studies

#### 2.6.1. Structure Preparation

A 3D structure for the SBE-β-CD was not available, therefore, the structure was built according to our precedent study [47]. The 3D coordinates of the RTN molecule were downloaded from Pubchem (https://pubchem.ncbi.nlm.nih.gov/compound/Rutin, accessed on 10 January 2024).

#### 2.6.2. Molecular Dynamics

The molecular dynamics (MD) simulation was made in explicit water using the YASARA Structure package (23.9.29) [48], according to previously reported procedures [49,50]. The RTN and SBE-β-CD force field parameters were generated using the GAFF2 [51] and AM1BCC [52] force fields as well as TIP3P for water. For Van der Waals forces (the default value used by AMBER), the cutoff was 10 Å [53], but no cutoff was used for electrostatic forces (the Particle Mesh Ewald method) [54]. Using procedures previously discussed in detail, the equations of movements were integrated with multiple time steps of 2.5 fs for bonded interactions and 5.0 fs for nonbonded interactions at a temperature of 298 K and a pressure of 1 atm (NPT ensemble) [55]. The finished system was around 30 × 36 × 32 Å3. To eliminate clashes, a brief MD simulation was performed solely on the solvent. Then, in order to remove conformational stress, the complete system was energy reduced using the steepest descent minimization. This was followed by a simulated annealing minimization until convergence (<0.01 kcal/mol Å). Ultimately, unrestricted MD simulations lasting 500 ns were run and the conformations of each system were recorded every 250 ps. The latest three ns averaged structures were taken into consideration for additional analysis after the solute RMSD as a function of simulation time was examined.

#### 2.6.3. Binding Free Energy Calculation

The binding free energy was computed using the well-known and extensively applied molecular mechanics Poisson–Boltzmann surface area (MM/PBSA) approach [56] on the optimized MD structure acquired from the previous phase. YASARA adopted Nunthaboot’s consolidated procedure for this process [57].

### 2.7. Wide-Angle X-ray Diffraction (WAXD) 

WAXD patterns of RTN/SBE-β-CD inclusion complex were performed in comparison with free components and the physical mixture by using an automatic Bruker D8 Advance diffractometer (Bruker, Billerica, MA, USA), with nickel-filtered Cu–Kα radiation. The analyses were performed in reflection, in 4°–90° as the 2θ range, being 2θ the peak diffraction angle.

### 2.8. Thermogravimetric Analysis (TGA)

Thermal properties of the inclusion complex in comparison with the free components and RTN/SBE-β-CD physical mixture were investigated by using a Perkin Elmer STA 6000 instrument (PerkinElmer Inc., Waltham, MA, USA). All scans were performed under nitrogen. A weighed amount of each sample (5–10 mg) was heated from 30 °C to 300 °C at a heating rate of 10 °C min^−1^, kept at 300 °C for 10 min, and then heated up to 550 °C at 20 °C min^−1^. Samples were maintained at 550 °C for 15 min, then further heated up to 900 °C (heating rate of 10 °C/min) and kept at 900 °C for 3 min, then kept at 900 °C for 30 min under flowing air (60 mL/min).

### 2.9. Fourier-Transform Infrared (FT-IR) Spectroscopy

The FT-IR spectra of the inclusion complex, free components and the physical mixture were performed at 25 °C, with a wavenumber ranging from 4000 cm^−1^ to 400 cm^−1^; this was done using a FT-IR Nicolet iS5 spectrometer (Nicolet Instrument Corporation, Madison, WI, USA).

### 2.10. Scanning Electron Microscopy (SEM)

The surface morphologies of free RTN, SBE-β-CD, the physical mixture, and their inclusion complex were investigated through scanning electron microscopy. SEM images were acquired using a Zeiss Evo 50 EP (Carl Zeiss SMT, Oberkochen, Germany) with an operating voltage of 15 kV. All samples were gold-coated using a sputtering system before imaging; a coating of roughly 10 nm was deposited. 

### 2.11. Determination of Water Solubility and Dissolution Profile of RTN/SBE-β-CD Inclusion Complex

The solubility of RTN/SBE-β-CD inclusion complex was determined in the dark at 25.0 ± 0.5 °C. An excess of freeze-dried RTN/SBE-β-CD inclusion complex was dispersed into 10 mL of water. The suspension was kept under magnetic stirring for 24 h until the equilibrium was reached. At timed intervals a sample of suspension was collected, filtered and the RTN quantified by UV-vis spectroscopy (diode array spectrophotometer, StellarNet BLACK-Comet, Model C—Tampa, FL, USA).

Following the 44th United States Pharmacopoeia (USP), paddle method dissolution studies were conducted in the dark at 37.0 ± 0.5 °C, keeping the dissolution medium in a constant, smooth motion (100 rpm). Free RTN (150 mg) or a corresponding amount in the complex was suspended in 900 mL of water. At fixed times (15, 30, 45, 60, and 120 min), 1 mL aliquot was collected, filtered, and analyzed by UV-vis using a diode array spectrophotometer StellarNet BLACK-Comet, Model C (Tampa, FL, USA) to determine the concentration of RTN in the solution. The volume was kept constant by adding fresh preheated water, and the data were corrected according to the dilutions. Experiments were conducted in triplicate, and sink conditions were maintained for all the experiments.

### 2.12. In Vitro Antibacterial and Antibiofilm Activity

#### 2.12.1. Strains

The following strains were used in this study: *Staphylococcus aureus* ATCC 6538, biofilm-producing reference strain [58], *S. aureus* ATCC 43300 (Methicillin resistant *S. aureus*-MRSA), *Pseudomonas aeruginosa* ATCC 9027, *P. aeruginosa* DSM 102273 (multidrug resistant). The strains were stored in the private collection of the Department of Chemical, Biological, Pharmaceutical and Environmental Sciences, University of Messina (Messina, Italy). They were stored at −70 °C in Microbanks™ (Pro-lab Diagnostics, Neston, UK). 

#### 2.12.2. Susceptibility Tests

The antibacterial activity of the RTN/SBE-β-CD inclusion complex was evaluated compared with free RTN and SBE-β-CD by determining the minimum inhibitory concentration (MIC) and the minimum bactericidal concentration (MBC) against the strains mentioned above. The MIC was assessed using the broth microdilution method, following the Clinical and Laboratory Standards Institute, with some modifications [59]. An overnight culture in Müeller–Hinton broth for each strain was adjusted to the required inoculum of 1 × 10^6^ enumerate colony-forming units (CFU)/mL. Aliquots of 100 μL of each suspension were inoculated in a 96-well microtiter plate containing a serial 2-fold dilution of free RTN, free SBE-β-CD, RTN/SBE-β-CD inclusion complex or 100 μL medium (growth control). Negative controls (medium + samples) were included. Levofloxacin hydrochloride (LVF) was used as the positive drug control. The bacterial growth was indicated visually and by a developer of enzymatic activity (Triphenyl tetrazolium chloride 0.05%, TTC). This revealed bacterial growth, which showed as a purple color after 15 min heating at 37 °C. The maximum RTN concentration tested was 1250 μg/mL as soluble RTN/SBE-β-CD inclusion complex, 150 μg/mL for free RTN (maximum solubility) and 8875 μg/mL for free SBE-β-CD, corresponding to the amount present in the assayed inclusion complex. The MIC-values, representing the lowest concentration showing no visible bacterial growth, were obtained after 24 h of incubation at 37 ± 1 °C. The MBC was evaluated by pipetting 10 μL from each clear well onto Müeller–Hinton agar plates. After 24 h of incubation at 37 °C, MBC was read as the lowest concentration which resulted in killing 99.9% of the initial inoculum. All experiments were repeated in triplicate.

#### 2.12.3. Effect on Biofilm Biomass and Viability

The antibiofilm effect was assessed, as described by Cramton et al. [60], with some modifications. Overnight cultures in Tryptic Soy Broth (TSB) with 1% glucose (TSBG) or without were diluted to standardize *S. aureus* or *P. aeruginosa* suspensions (1 × 10^6^ CFU/mL), respectively. Aliquots of 100 μL were dispensed into each well of sterile flat-bottom 96-well polystyrene microtiter plates (Corning Inc., Corning, New York, NY, USA) in the presence of samples at sub-MIC concentrations (0.5 MIC) or 100 μL medium (positive control). Negative controls (medium + samples) were included. The microtiter plates were incubated for 24 h at 37 °C. The medium was then aspirated and the wells were rinsed twice with phosphate-buffered saline (PBS), and fixed by drying for 1 h. Once the wells were fully dry, 200 μL of 0.1% safranin was added for 5 min. The contents of the wells were then aspirated and, after rinsing with water, 200 μL of 30% acetic acid (*v*/*v*) was added to the wells for spectrophotometric analysis. The OD 492 nm value obtained for each strain without a sample was used as the control. The reduction percentage of biofilm biomass formation in the presence of different samples was calculated using the ratio between the values of OD 492 nm with and without samples, adopting the following formula: [(OD control − OD sample)/OD control) × 100]

All experiments were repeated in triplicate. At the same time, the percentage of viable biomass was determined by the CFU counting method. Biofilm bacteria were scraped thoroughly from the wells with a pipet tip, with particular attention to the edges. The bacterial suspensions obtained were serially 10-fold diluted, plated on Tryptic Soy Agar plates, and grown for 24 h at 37 °C to enumerate CFU. Viability measures are derived from three separate experiments.

### 2.13. Statistical Analysis

Each measurement was repeated three times and the results were expressed as mean ± standard deviation (SD). Values were processed via one-way and two-way analysis of variance (ANOVA) followed by a Bonferroni post hoc test for multiple comparisons. A value of *p* < 0.05 was significant.

## 3. Results and Discussion

### 3.1. Studies of RTN/SBE-β-CD Characteristics

The interaction between SBE-β-CD and RTN has been extensively studied in solution and at solid state. RTN/SBE-β-CD inclusion complex was prepared by freeze-drying a hydroalcoholic solution containing RTN and SBE-β-CD in a 1:2 molar ratio. Twenty percent (*v*/*v*) of MeOH was added to allow complete solubilization of RTN, favoring the complexation. The excess of SBE-β-CD to 1:1 molar ratio was used to maintain the complex in solution.

#### 3.1.1. UV-Vis Spectroscopy Studies

The host–guest interaction between RTN and SBE-β-CD was studied by UV-vis spectroscopy. In Figure 2, we showed the spectra of free RTN and in the presence of increasing concentrations of SBE-β-CD. RTN shows two very intense bands; the first at 256 nm (Band II) and the second at 351 nm (Band I) [32]. These are both attributed to the π→π* transitions. Mabri et al. [61] attributed the band at 256 nm to the π→π* electronic rearrangement of the phenyl group, and the band at 351 nm to the benzene ring of the pyrocatecholic moiety, which can be regarded as an acyl-disubstituted benzene chromophore. Increasing amounts of SBE-β-CD resulted in a significant variation of the UV-vis spectrum of RTN. Both absorption bands shifted progressively towards the blue (hypsochromic effect) and increased in intensity (Figure 2). The significant variation of the RTN spectrum observed in the presence of SBE-β-CD and the high value of the apparent association constant highlight variation of the local polarity of the RTN chromophores which pass from the hydrophilic environment into the apolar cavity of the macrocycle, with a corresponding perturbation of the electronic transition. Furthermore, we cannot exclude that complexation produces the breakdown of the intramolecular hydrogen bonds present in the RTN molecule between the C3-OH oxygen atom with the C4-OH proton of the A ring and between the C3-OH proton of the A ring with the rhamnosidic C2-OH oxygen atom [62], followed by an establishment of other hydrogen bonds between pyrocatecholic moiety of RTN and the sulfobutyl moiety of the CD.

The binding stoichiometries were analyzed by using SupraFit software (2.5.120) [63]. The results of the UV-vis titration were analyzed using a 1:1, 2:1/1:1, 1:1/1:2, and a 2:1/1:1/1:2 model as implemented in SupraFit. Each best-fit model was inspected using Monte Carlo simulation to identify the 1:1/1:2 (RTN/SBE-β-CD) as the best-fitted model with a K_1:1_ of 32267 M^−1^ and a K_1:2_ of 12 M^−1^ with an associated DG_binding_ of −6.6 and −1.5 kcal/mol. Effectively, the almost exclusive complex in solution is the 1:1 one, whereas the 1:2 is present in a small amount at a high concentration of SBE-β-CD.

#### 3.1.2. Phase Solubility Studies

Figure 3 shows the phase solubility diagram obtained for RTN in the presence of increasing concentrations of SBE-β-CD. The graph obtained representing RTN concentration vs. SBE-β-CD concentration is of A_L_ type, showing that RTN/SBE-β-CD interaction leads to a soluble complex in the range of macrocycle concentrations considered. The slope of the graph is less than one, demonstrating the formation of a complex with 1:1 stoichiometry, and the association constant (K_c_) determined following Higuchi and Connors was 9660 M^−1^. This result is in line with UV-vis titration, which employs a more accurate method.

#### 3.1.3. NMR Investigation

Nuclear magnetic resonance (NMR) is a fundamental spectroscopic technique to investigate the formation and geometry of inclusion complexes. The chemical and electronic environments of the protons are modified by the host–guest interactions; therefore, during the complexation, a chemically induced shift of the corresponding protons has been observed. Unfortunately, like most substituted CDs, SBE-β-CD is a statistical mixture of the different stereoisomers, with broad unresolved peaks; this makes it almost impossible to follow the chemical shifts of its protons, especially the H-3 and H-5 protons protruding into the CD cavity, even though these protons were identified through 2D COSY spectra [64]. All the RTN protons of the flavonoid moiety displayed chemical shifts between 6.30 and 7.67 ppm, which are free and more evident than the broad and unsolved peaks of SBE-β-CD proton (mainly at δ 3–4 ppm). Therefore, the RTN/SBE-β-CD inclusion complex formation was deduced from the chemical shift changes observed in ^1^H NMR of the RTN aromatic protons previously measured in the free state. The enumeration of the RTN structure is shown in Figure 4, together with the stacked portions of the ^1^H NMR spectra of RTN and RTN/SBE-β-CD inclusion complex. In Table 1, we reported the chemical shift of free and complexed RTN (RTN/SBE-β-CD 1:1 molar ratio). 

The chemical shift changes of the RTN aromatic protons are diagnostics. The H-6 and H-8 protons of the RTN A ring resonate at 6.30 and 6.49 ppm, while H-2′, H-5′, and H-6′ B ring protons have chemical shifts at 7.67, 6.98, and 7.61 ppm, respectively. The inclusion of RTN in the SBE-β-CD hydrophobic cavity was confirmed by changes in the chemical shifts of the guest and host protons that were observed in the RTN/SBE-β-CD inclusion complex spectrum, in comparison with the chemical shift observed for the same protons in the free RTN.

In the RTN/SBE-β-CD spectrum of the complex (performed in a 7:3 D_2_O/CD_3_OD solution), significant changes were observed in the chemical shifts of H-8 RTN proton (∆δ 1.07) due to the proximity to the oxygen atom of the 1,4-glycosidic bonds between the seven glucopyranose units which constitute the SBE-β-CD molecule. Other interesting changes were observed for RTN protons H-6, H-2′, and H–5′ in the RTN/SBE-β-CD complex spectrum (Table 1). From these results, it can be assumed that the RTN benzopyranone skeleton was incorporated in the SBE-β-CD hydrophobic cavity (Figure 5).

The changes observed for H-2′, H-5′, and H–6′ protons of the RTN ring B indicate that this latter is probably in close contact with the sulfobutylether groups [19].

#### 3.1.4. Molecular Modeling Studies

In order to investigate the RTN/SBE-β-CD host–guest interactions, we started the molecular modeling study docking the RTN into the SBE-β-CD hydrophobic cavity, as described in the Appendix A. From the 10 most stable docked poses, the two that corresponded to the orientation of ring A or B within the hydrophobic cavity from the secondary edge were selected. These two complexes were subjected to a 500 ns molecular dynamics simulation. The analysis of the two simulations showed that after about 100 ns the molecule of RTN is stabilized into the SBE-β-CD cavity. Once inside the host molecule, the RTN establishes interactions that stabilize the complex and for the entire simulation time (500 ns) stays inside the CD, maintaining the same orientation. On all the trajectories of the MD simulation, we performed an MM/PBSA calculation to obtain the binding energy and, consequently, to select the most stable complex structures for both A and B ring orientations. Of these two optimized structures, ring A inside the RTN/SBE-β-CD hydrophobic cavity resulted in the most stability. This last structure, reported in Figure 5, is consistent with the complex geometry deduced by the NMR experiments. The A ring is fully inserted into the cavity and justifies the high shift registered for the H-8; even the C ring is partially inserted into the cavity, in line with the H2′ and H6′ shifts. Finally, the methyl group in one of the sugar rings is located between two of the seven sulfonic groups to deshield it; this also agrees with the magnitude of the low-field shift observed in NMR analysis. This structure is stabilized by two hydrogen bonds between two hydroxyl moieties of the sugar rings that interact with one of the alcoholic oxygens of the secondary rim and with one of the sulphonic oxygens, respectively.

#### 3.1.5. WAXD, TGA, and FT-IR Analyses

The inclusion complex was characterized at solid state by X-ray diffractometry (WAXD), thermogravimetric analysis (TGA), and Fourier transform infrared spectroscopy (FT-IR).

WAXD patterns obtained for RTN/SBE-β-CD inclusion complex, free components, and RTN/SBE-β-CD physical mixture are shown in Figure 6. RTN shows an intense and sharp peak that evidences its crystalline nature. They were still present in the physical mixture but disappeared in the inclusion complex’s spectrum. This latter showed a broad and diffuse signal without sharp crystalline peaks, indicating the amorphous nature of the complex due to the drug’s inclusion within the SBE-β-CD cavity.

The thermal properties of the RTN/SBE-β-CD inclusion complex, free RTN, SBE-β-CD, and the physical mixture were evaluated by TGA (Figure 7). For SBE-β-CD, the mass loss below 150 °C can be attributed to surface water evaporation associated with the macrocycle. For the RTN/SBE-β-CD inclusion complex, the mass loss in the 150–550 °C range may be related to internal water evaporation and SBE-β-CD or drug degradation. Furthermore, a slight modification of the degradation temperature of the macrocycle in the inclusion complex is highlighted, which would indeed suggest the formation of the inclusion complex in the solid phase. As can be seen, the thermal stability of RTN increases when complexed with SBE-β-CD.

Finally, FT-IR spectroscopy was used to verify the complexation. The variation of intensity, changes in wavenumbers, disappearance and/or magnification of the typical FT-IR bands of the functional groups of the molecules suggest the formation of an inclusion complex [65]. The FT-IR spectra recorded at 25 °C of RTN/SBE-β-CD inclusion complex, free RTN and SBE-β-CD, and of the physical mixture are shown in Figure 8. The RTN spectrum is characterized by stretching vibrations of the hydroxyl groups, and intense bands are observable for the vibrations of the C=O, C=C, C-O, and C-O-C groups. Instead, the spectrum of SBE-β-CD has absorption bands for the hydroxyl groups, -CH and -CH_2_, and for the C-O-C stretching vibration. In the inclusion complex, some IR drug bands are absent or reduced in intensity, suggesting that RTN is trapped in the macrocycle cavity.

#### 3.1.6. Scanning Electron Microscopy (SEM) Analysis

SEM is used in the solid-state characterization of the raw materials, and their corresponding physical mixtures and inclusion complexes [65]. This technique is inadequate to identify the formation mechanism of the complex, but it evidences morphological changes related to the interactions between the components [66]. The surface morphologies of free RTN, SBE-β-CD, the physical mixture, and the inclusion complex are shown in Figure 9a–d. Free RTN appeared with crystals of different sizes in rectangular blocks (Figure 9a), while the SBE-β-CD had an almost circular form with a perfectly smooth surface (Figure 9b). Both RTN and SBE-β-CD particles were still evident in the SEM image of the physical mixture. As shown in Figure 9c, the circular particles do not appear perfectly smooth, but a rough, irregular, and opaque surface characterizes this substrate. This different morphology suggests that the drug appears to adhere to the surface of SBE-β-CD in the physical mixture, slightly modifying its structure without interacting with it. On the other hand, the inclusion complex showed drastic changes in particle size and the shape of particles. The original morphology of the two pure compounds was lost, while a single solid structure is formed (Figure 9d). Thus, the very different shape and morphology can be considered as proof that a new structure was present.

### 3.2. Water Solubility and Dissolution Profile of RTN/SBE-β-CD Inclusion Complex

The inclusion of RTN into SBE-β-CD resulted in an approximately 20-fold increase in water solubility compared to the free drug (0.125 mg/mL in water) [48]. Due to the high solubility and complexing ability of the CD used in this study, the solubility increase of RTN is higher than that reported by other authors in the presence of different CDs [35,36]. Complete dissolution of the complex was observed in about 30 min, whereas only 10% of free RTN was dissolved in the same period of time and under the same conditions (Figure 10).

### 3.3. Antibacterial and Antibiofilm Activity 

#### 3.3.1. Bacterial Susceptibility

The antibacterial activity of the RTN/SBE-β-CD inclusion complex compared to free RTN and SBE-β-CD was assayed against *S. aureus* and *P. aeruginosa*, including resistant strains. The activity of the samples was assayed by minimum inhibitory concentration (MIC) and minimum bactericidal concentration (MBC) methods. Due to its minimal solubility, free RTN was assayed at 150 μg/mL; higher concentrations rapidly precipitated in the culture medium. Several authors demonstrated the antibacterial activity of free RTN, although different MIC values were reported on the same strains [35,67]. Among the possible reasons, the different MIC values obtained may be due to solubility issues leading to poorly defined concentrations. When complexed with the macrocycle, RTN significantly increases its water solubility; thus, a higher concentration of complexed RTN (1250 μg/mL) can be assayed to the free drug. As reported in Table 2, free RTN showed bacteriostatic activity against *S. aureus* ATCC 6538 and *P. aeruginosa* ATCC 9027 with MIC values of 75 and 150 μg/mL, respectively. No activity was observed against resistant strains. As demonstrated by other authors, the RTN activity could be due to several mechanisms such as damage to the bacterial cell membrane [4] or inhibition of nucleic acid synthesis [68], efflux pumps [69], toxin formation and biofilm formation [70]. Free SBE-β-CD showed no activity against all the tested strains. Interesting results were obtained for the RTN/SBE-β-CD inclusion complex. The presence of macrocycle increased RTN effectiveness against all tested strains. Complexed RTN showed bactericidal activity against *S. aureus* ATCC 6538 and ATCC 43300 (MRSA) at a concentration of 4.88 μg/mL and against *P. aeruginosa* ATCC 9027 at 39.06 μg/mL. Furthermore, it showed bacteriostatic activity against *P. aeruginosa* DSM 102273 resistant strain with a MIC value of 1250 μg/mL. These results could depend not only on the increased solubility of RTN by the complexation, but also on the permeabilization of the bacterial wall produced by the macrocycle [28,71]. 

#### 3.3.2. Antibiofilm Effect

Biofilm is produced by microorganisms as an extracellular matrix and is composed of polysaccharides, secreted proteins, and extracellular DNA. It is considered an important virulence factor causing chronic infections [72]. *S. aureus* and *P. aeruginosa* are the most common bacterial pathogens associated with different types of eye surface infections including keratitis, dacryocystitis, blepharokeratoconjunctivitis, and conjunctivitis [11]. Chronic *S. aureus* and *P. aeruginosa* ocular infections associated with biofilms have become increasingly challenging to treat with current antimicrobials [11,73,74].

The activity of the RTN/SBE-β-CD inclusion complex compared to free RTN and SBE-β-CD against biofilm formation was measured by the crystal violet method and CFU assay. Free RTN and free SBE-β-CD showed no antibiofilm effect against all strains at 0.5 MIC concentration. The inclusion complex significantly decreased biofilm biomass production at 0.5 MIC against *S. aureus* ATCC 6538 (biofilm reduction 57%) and *S. aureus* MRSA (biofilm reduction 41%). The percentage of inhibition was found to be statistically significant compared with the negative control at *p* < 0.05 for *S. aureus* ATCC 6538 and at *p* < 0.01 for *S. aureus* MRSA (Figure 11a and Figure 12a). 

Moreover, at 0.5 MIC the inclusion complex reduced the viability of *S. aureus* ATCC 6538 by 2 log_10_ units (*p* < 0.0001), and that of *S. aureus* MRSA by 3.6 log_10_ units (*p* < 0.0001) (Figure 11b and Figure 12b).

**Figure 12 pharmaceutics-16-00233-f012:**
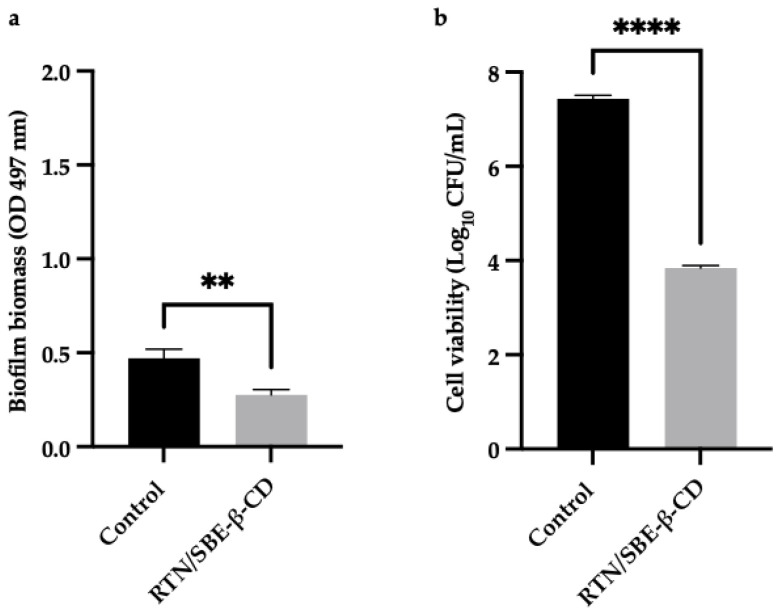
The results are expressed as mean ± SD. RTN/SBE-β-CD inclusion complex significantly reduced *S. aureus* ATCC 43300 (MRSA) biofilm formation at 0.5 MIC (0.61 μg/mL in RTN). (**a**) Biofilm biomass is expressed as crystal violet optical density (O.D. 497 nm) (** *p* < 0.01); (**b**) cell viability is expressed as Log_10_ CFU/mL (**** *p* < 0.0001).

However, RTN/SBE-β-CD inclusion complex showed no inhibitory effect on biofilm production against *P. aeruginosa* strains at 0.5 MIC, which is equivalent to 0.25 μg/mL and 62.5 μg/mL for *P. aeruginosa* ATCC 9027 and *P. aeruginosa* DSM 102273, respectively.

## 4. Conclusions

In this study, we demonstrated the ability of SBE-β-CD to complex the natural drug RTN to improve its water solubility and dissolution and potentiate its antibacterial effect. As in-solution and solid-state investigations demonstrated, a stable inclusion complex in a 1:1 molar ratio was formed. The complexation potentiates the antibacterial activity of RTN against Gram-positive and Gram-negative strains. Remarkably, the MIC and MBC of complexed RTN are, respectively, 60 times and 30 times higher than the free drug against *S. aureus* ATCC 6538. The complexed RTN shows similar high activity against *S. aureus* ATCC 43300 (MRSA), while free RTN shows no activity. The macrocycle increases the antibacterial activity of RTN, albeit less intense, towards the Gram-negative *P. aeruginosa* ATCC 9027. Furthermore, the inclusion complex has significant antibiofilm activity against *S. aureus* strains, reducing viability cells; while no antibiofilm activity was detected for free RTN.

Our results suggest that SBE-β-CD could be a suitable carrier for RTN, permitting the realization of liquid formulation for ophthalmic administration with antibacterial and antibiofilm properties, and representing a good starting point for our successive in vitro and in vivo investigations.

## Figures and Tables

**Figure 1 pharmaceutics-16-00233-f001:**
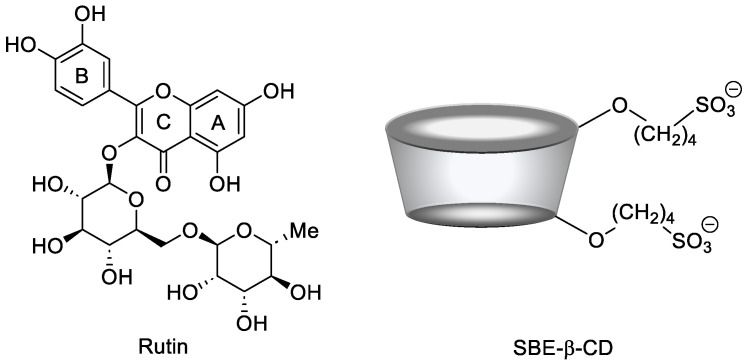
Molecular structure of RTN (**left**) and schematic structure of SBE-β-CD (**right**).

**Figure 2 pharmaceutics-16-00233-f002:**
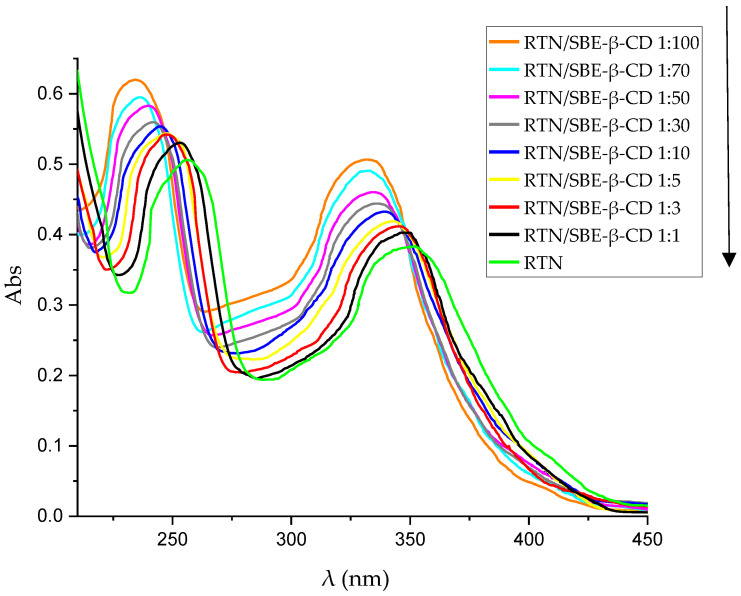
UV-vis spectra of free RTN and in the presence of an increasing amount of SBE-β-CD in water:MeOH 80:20 mixture (%, *v*/*v*). The experiments were carried out in triplicate.

**Figure 3 pharmaceutics-16-00233-f003:**
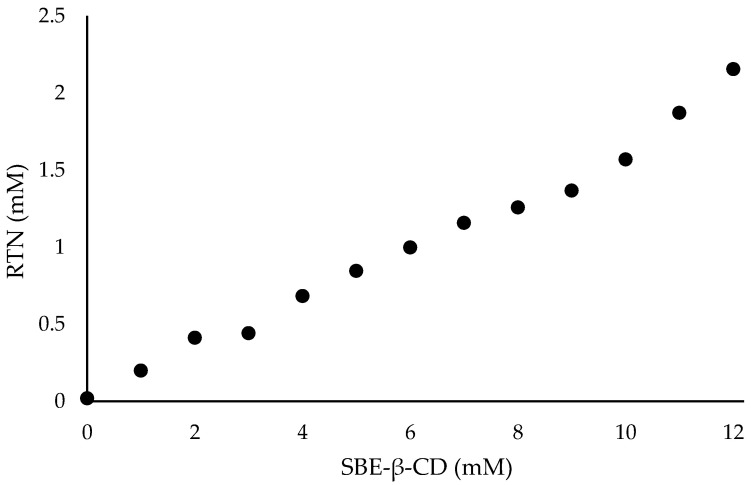
Phase solubility diagram of RTN/SBE-β-CD inclusion complex. The experiments were carried out in triplicate.

**Figure 4 pharmaceutics-16-00233-f004:**
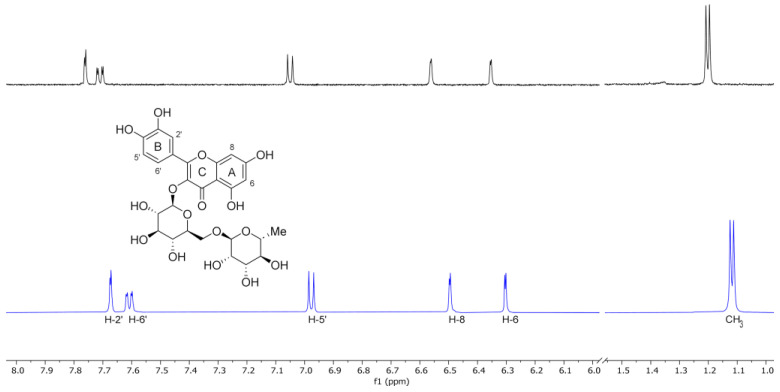
Stacked portions of the ^1^H NMR spectra relative to the free RTN (blue line) and RTN/SBE-β-CD (black line) inclusion complex. Only those diagnostic signals relative to RTN are reported.

**Figure 5 pharmaceutics-16-00233-f005:**
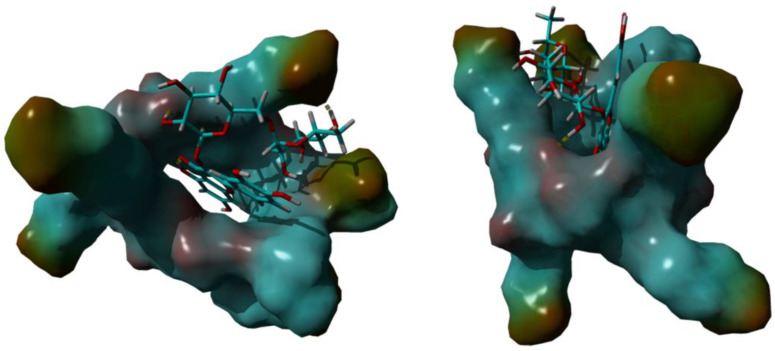
The 3D full minimized structure of RTN/SBE-β-CD inclusion complex. Top view (**left**) and side view (**right**). Hydrogen bonds were represented as a yellow dotted line.

**Figure 6 pharmaceutics-16-00233-f006:**
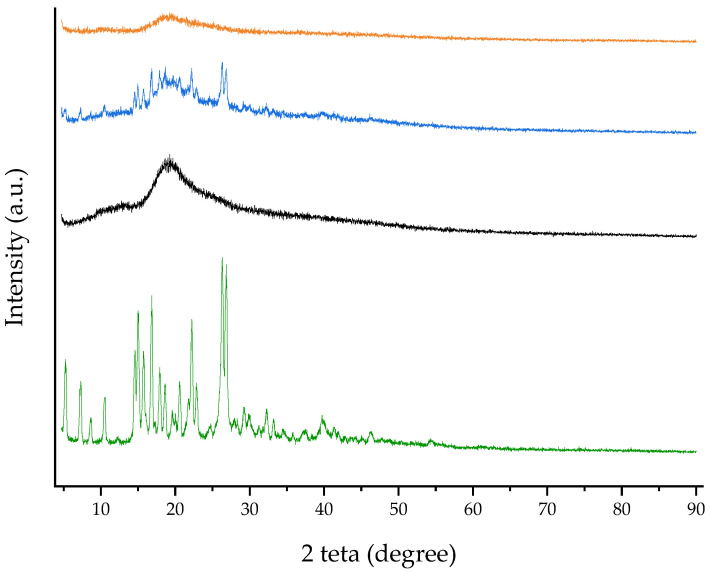
WAXD patterns of free RTN (green line), SBE-β-CD (black line), physical mixture (blue line), and inclusion complex (orange line).

**Figure 7 pharmaceutics-16-00233-f007:**
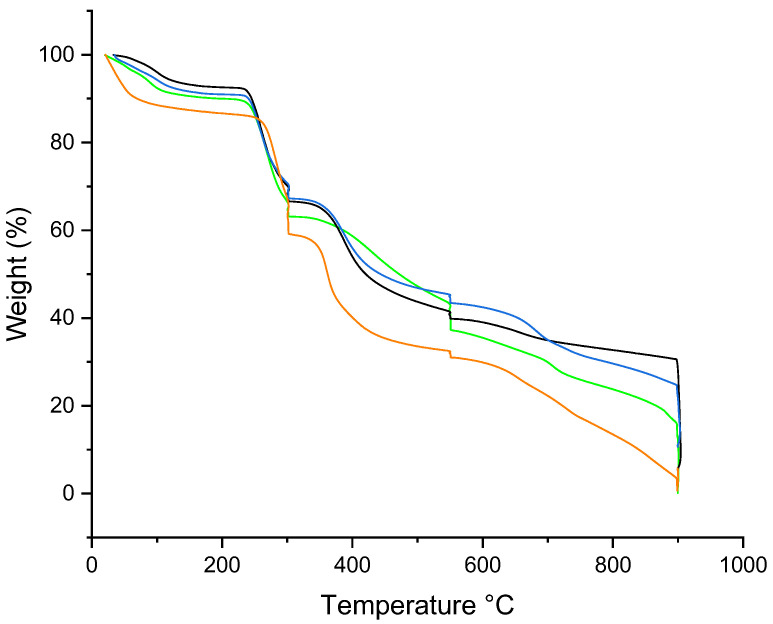
TGA curves of free RTN (green line), SBE-β-CD (black line), physical mixture (blue line), and inclusion complex (orange line).

**Figure 8 pharmaceutics-16-00233-f008:**
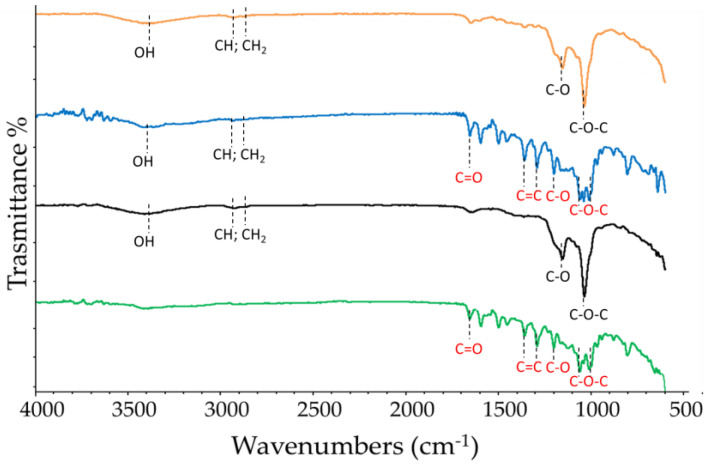
FT-IR spectra recorded at 25 °C of free RTN (green line), SBE-β-CD (black line) physical mixture (blue line), and inclusion complex (orange line). Characteristic peaks were labeled; the absorptions of RTN and SBE-β-CD functional groups are in red and black, respectively.

**Figure 9 pharmaceutics-16-00233-f009:**
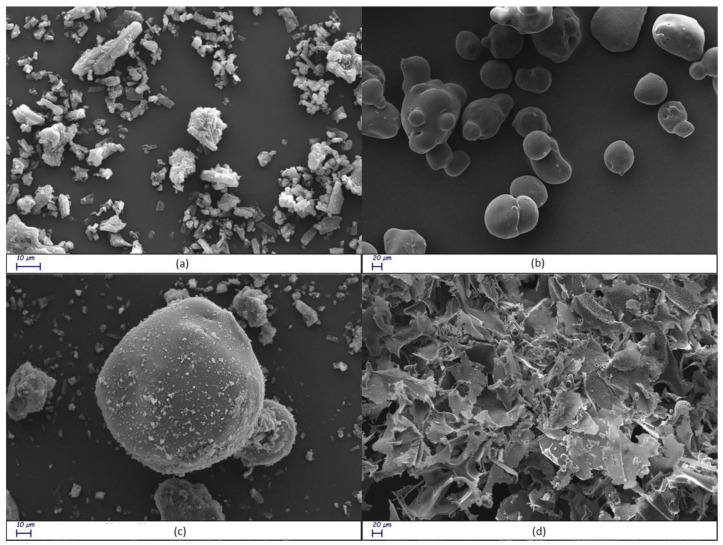
Scanning electron microscope (SEM) images of free RTN (**a**), SBE-β-CD (**b**), physical mixture (**c**), and inclusion complex (**d**) at different magnifications.

**Figure 10 pharmaceutics-16-00233-f010:**
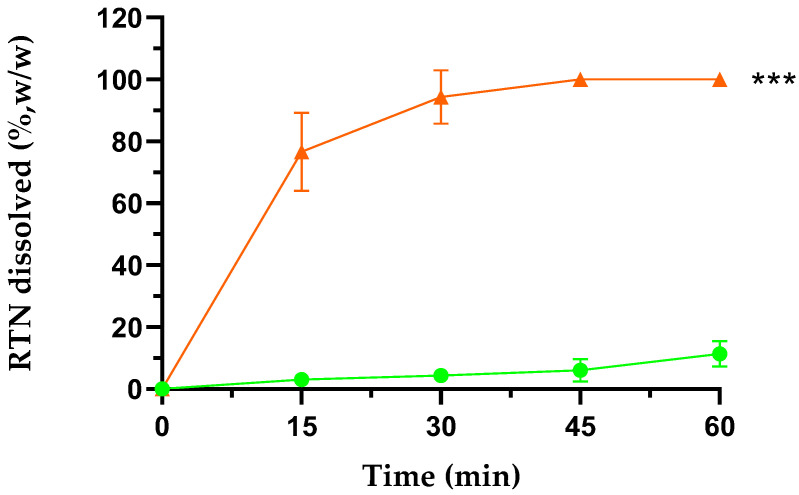
Dissolution profiles of free RTN (green line) and RTN/SBE-β-CD inclusion complex (orange line) in water at 25.0 ± 0.5 °C. All values of the inclusion complex are statistically significant compared to free RTN data (*** *p* < 0.001). The experiments were carried out in triplicate. The results are presented as the mean of three experiments ± standard deviation (SD). The error bar, if not shown, is inside the symbol.

**Figure 11 pharmaceutics-16-00233-f011:**
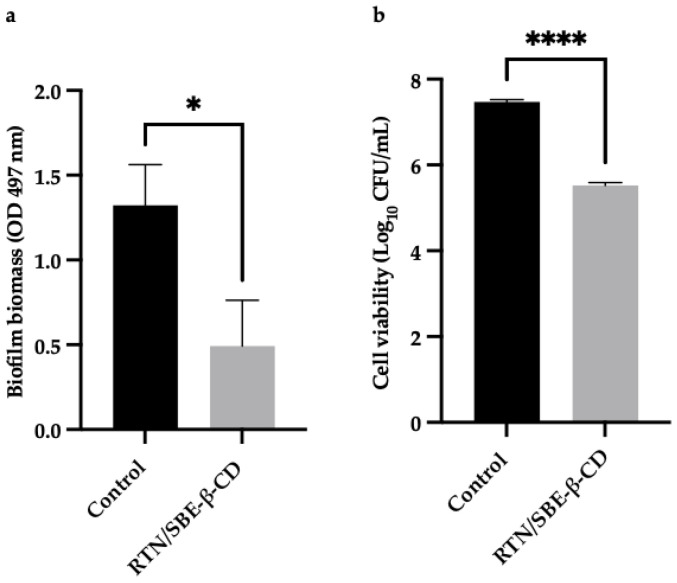
The results are expressed as mean ± SD. RTN/SBE-β-CD inclusion complex significantly reduced *S. aureus* ATCC 6538 biofilm formation at 0.5 MIC (0.61 μg/mL in RTN). (**a**) Biofilm biomass is expressed as crystal violet optical density (O.D. 497 nm) (* *p* < 0.05); (**b**) cell viability is expressed as Log_10_ CFU/mL (**** *p* < 0.0001).

**Table 1 pharmaceutics-16-00233-t001:** ^1^H NMR chemical shifts in δ and ∆δ of RTN protons in the free state and RTN/SBE-β-CD 1:1 complex [9 mM in a 7:3 D_2_O/MeOD solution]; for doublet or double-doublet, the reported δ refer to the centered signal.

Protons	RTN	RTN/SBE-β-CD	∆δ ^a^
2′	7.67 (d)	7.76 (d)	0.09
5′	6.98 (d)	7.05 (d)	0.07
6′	7.61 (dd)	7.71 (dd)	0.10
6	6.30 (d)	6.35 (d)	0.05
8	6.49 (d)	6.56 (d)	1.07
CH_3_	1.12 (d)	1.20 (d)	0.08

^a^ ∆δ = δ_complex_ − δ_free_.

**Table 2 pharmaceutics-16-00233-t002:** Minimum inhibitory concentration (MIC) and minimum bactericidal concentration (MBC) of RTN/SBE-β-CD inclusion complex compared to free RTN.

Strains	Free RTN (μg/mL)	RTN/SBE-β-CD ^a^ (μg/mL)	LVF
*S. aureus* ATCC 6538			
MIC	75	1.22	0.5
MBC	150	4.88	1
*S. aureus* ATCC 43300			
MIC	—	1.22	0.008
MBC	—	4.88	0.031
*P. aeruginosa* ATCC 9027			
MIC	150	39.06	2
MBC	—	39.06	62.5
*P. aeruginosa* DSM 102273			
MIC	—	1250	125
MBC	—	—	500

^a^ The concentration is referred to as the complexed RTN; —no activity. LVF (drug control).

## Data Availability

Data are contained within the article.

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
