# Peer review of "Rutin/Sulfobutylether-β-Cyclodextrin as a Promising Therapeutic Formulation for Ocular Infection"

_pharmaceutics, 2024, doi:10.3390/pharmaceutics16020233_

Round 1

Reviewer 1 Report

Comments and Suggestions for Authors

This manuscript presents the development and characterization of a pharmaceutical system formed by the complexation of rutin (glycosylated flavonoid) with the excipient Captisol®, to increase the solubility of the API.

Captisol is a sulfobutylether β-cyclodextrin (SBE-β-CD) known for its ability to form complexes with drugs, increasing their solubility and also has low toxicity. SBE-β-CD has been the subject of numerous research articles, with the first works and patents appearing in the last decade of the 20th century.

Currently there are 15 medications approved by the FDA for use in clinical therapy that contain Captisol as an excipient. The first drug with this formulation was approved in 2001 (ziprasidone mesylate for intramuscular injection).

The authors have extensive experience with CD, having published their studies in journals such as Carbohydrate Polymers 2019, Biomelucules 2022, Pharmaceutics and Molecules 2023, etc., that have been cited in the bibliography of the manuscript.

The methodology is also used by other authors for the physicochemical characterization of CD-drug complexes. An example of this is the following articles, whose content is similar to that of the manuscript submitted for review.

2011 (doi: 10.1208/s12249-011-9685-z), Sulfobutyl Ether7 β-Cyclodextrin (SBE7 β-CD) Carbamazepine Complex: Preparation, Characterization, Molecular Modeling, and Evaluation of In Vivo Anti-epileptic Activity, de Ankitkumar S. Jain et al.

2023 (doi: 10.3390/molecules28030955) Preparation, Characterization and Molecular Dynamics Simulation of Rutin-Cyclodextrin Inclusion Complexes de Chang C. et al

It would be advisable to include more information regarding SBE-β-CD in the introduction section, as well as when discussing the results.

Although the use of SBE-β-CD is not new, routine complexation is new in order to develop a pharmaceutical system for ocular application. Thus, the results achieved regarding antibacterial and antibiofilm activity are very promising

Author Response

Responses to Reviewer # 1

This manuscript presents the development and characterization of a pharmaceutical system formed by the complexation of rutin (glycosylated flavonoid) with the excipient Captisol®, to increase the solubility of the API.

Captisol is a sulfobutylether β-cyclodextrin (SBE-β-CD) known for its ability to form complexes with drugs, increasing their solubility and also has low toxicity. SBE-β-CD has been the subject of numerous research articles, with the first works and patents appearing in the last decade of the 20th century.

Currently there are 15 medications approved by the FDA for use in clinical therapy that contain Captisol as an excipient. The first drug with this formulation was approved in 2001 (ziprasidone mesylate for intramuscular injection).

The authors have extensive experience with CD, having published their studies in journals such as Carbohydrate Polymers 2019, Biomelucules 2022, Pharmaceutics and Molecules 2023, etc., that have been cited in the bibliography of the manuscript.

The methodology is also used by other authors for the physicochemical characterization of CD-drug complexes. An example of this is the following articles, whose content is similar to that of the manuscript submitted for review.

2011 (doi: 10.1208/s12249-011-9685-z), Sulfobutyl Ether7 β-Cyclodextrin (SBE7 β-CD) Carbamazepine Complex: Preparation, Characterization, Molecular Modeling, and Evaluation of In Vivo Anti-epileptic Activity, de Ankitkumar S. Jain et al.

2023 (doi: 10.3390/molecules28030955) Preparation, Characterization and Molecular Dynamics Simulation of Rutin-Cyclodextrin Inclusion Complexes de Chang C. et al

It would be advisable to include more information regarding SBE-β-CD in the introduction section, as well as when discussing the results.

Although the use of SBE-β-CD is not new, routine complexation is new in order to develop a pharmaceutical system for ocular application. Thus, the results achieved regarding antibacterial and antibiofilm activity are very promising

The authors are very grateful to the Reviewer for his/her positive judgement of the paper and for his/her suggestion to improve the quality of the work.

Below we reported the responses to the Reviewer’s comments. All changes made in the revised version of the manuscript are highlighted in yellow.

It would be advisable to include more information regarding SBE-β-CD in the introduction section, as well as when discussing the results.

As suggested by the Reviewer, more information concerning SBE-β-CD has been reported in the revised version of the manuscript (see lines 106-118).

Reviewer 2 Report

Comments and Suggestions for Authors

The topic of this manuscript is important and current, and results could be interesting for readers. The work is well planned and written with the support of current literature. However, some changes have to be entered into the revised version of the manuscript before it can be further processed:

1.     The introduction lacks a literature review on the combinations of rutin and cyclodextrins. And there are a lot of such reports. Please complete this.

2.     there is no explanation why the authors chose SBE to prepare the complex

3.     what is the advantage of the presented work compared to already published works? e.g. https://doi.org/10.4028/www.scientific.net/AMR.455-456.1177

4.     Chapter 2.3. - did the authors put the system in a capsule? How did the authors avoid flotation of the system on the liquid surface?

5.     Chapter 2.12.1 - strain names e.g. Staphylococcus aureus should be written in italic

6.     the results should be arranged in a logical manner, first preparing the system, then spectral and thermal characteristics, and only then assessing parameters such as release or microbiological activity

7.     the results of levofloxacin activity as a positive test should be presented in the manuscript

Author Response

Responses to Reviewer # 2

The topic of this manuscript is important and current, and results could be interesting for readers. The work is well planned and written with the support of current literature. However, some changes have to be entered into the revised version of the manuscript before it can be further processed:

 The authors are very grateful to the Reviewer for his/her positive judgement of the paper and for his/her suggestions to improve the quality of the work.

Below we report the responses to the Reviewer’s comments. All changes made in the revised version of the manuscript are highlighted in yellow.

  1. The introduction lacks a literature review on the combinations of rutin and cyclodextrins. And there are a lot of such reports. Please complete this.

In the revised version of the manuscript, we have extended the introduction concerning RTN/CD inclusion complexes, reporting some of the papers present in literature (see lines 85-105)

  1. there is no explanation why the authors chose SBE to prepare the complex

SBE-β-CD was selected for its higher water solubility and complexing ability with respect to native β-CD or other modified CDs usually present in pharmaceutical products. It shows high biocompatibility, non-toxicity and it is approved by FDA for parenteral administration. Fifteen products are on the market today containing SBE-β-CD.  Based on excellent properties showed by this macrocycle, a greater improvement of physical-chemical and biological properties of RTN could be expected using SBE-β-CD rather than other CDs. Furthermore, a very substantial literature is present concerning the complexation of RTN with native β-CD and different modified β-CD, but, to the best of our knowledge, very few papers describe the physical-chemical properties of RTN/SBE-β-CD inclusion complex and none of these reported its antimicrobial and antibiofilm activity neither in vitro nor in vivo.

We have added this explanation in the text (see lines 107-119 of the revised manuscript)

  1. what is the advantage of the presented work compared to already published works? e.g. https://doi.org/10.4028/www.scientific.net/AMR.455-456.1177

The authors are grateful to the Reviewer for this observation.

The works present in literature concerning RTN/SBE-β-CD inclusion complex described the preparation of the complex and its physical-chemical characterization that, in the most cases, is limited to the determination of the association constant of the complex.

In particular, the work of Zhou et al. (inserted in the revised version of the manuscript as Reference n. 42) that the Reviewer mentions, describes different methods to obtain the complex and reports a sparse characterization of the complex by using UV-vis spectroscopy and phase-solubility study. No biological studies were performed.

In our work, we prepared RTN/SBE-β-CD inclusion complex and conducted a deep and full characterization of the complex at solid state and in solution, by using different technique, that is, Wide-angle X-ray diffraction, Thermogravimetric analysis, Fourier-transform infrared spectroscopy, UV-vis and NMR spectroscopy, molecular modelling, and morphology. Furthermore, the correct stoichiometry and the accurate binding constant were determined by employing the SupraFit software. Our work concludes with biological antibacterial and antibiofilm studies of RTN/SBE-β-CD inclusion complex, not present in previous published papers, that demonstrate higher activity of this complex with respect to the free drug. This probably due to the higher increase of RTN water solubility observed in the presence of SBE-β-CD than other CDs.

We have extended the text of the revised version of the manuscript (see lines 106-118), reporting new references (see references n 43-45).

  1. Chapter 2.3. - did the authors put the system in a capsule? How did the authors avoid flotation of the system on the liquid surface?

The inclusion complex shows excellent wettability; therefore, we had no problems of flotation.

  1. Chapter 2.12.1 - strain names e.g. Staphylococcus aureus should be written in italic

The text has been entirely checked to write the strain names in italic.

  1. the results should be arranged in a logical manner, first preparing the system, then spectral and thermal characteristics, and only then assessing parameters such as release or microbiological activity

As requested by the Reviewer, the authors have rearranged the position of the paragraphs concerning experimental section and results. Furthermore, we have added within the paragraph  2.11. the method used to determine the water solubility of the inclusion complex (see lines 240-245). As a consequence of the modifications, we have changed the numbers of paragraphs and of Figures.

  1. the results of levofloxacin activity as a positive test should be presented in the manuscript

As requested by the Reviewer, the results of levofloxacin activity, as a positive test, were shown in table 2 of the revised manuscript.

Reviewer 3 Report

Comments and Suggestions for Authors

The manuscript outlines a study on a new therapeutic formulation for ocular infections, using a rutin/sulfobutylether-β-cyclodextrin (RTN/SBE-β-CD) inclusion complex. The major aims were to improve the solubility and therapeutic efficacy of rutin, a natural flavonoid with potential antibacterial properties. The study found that the RTN/SBE-β-CD complex significantly enhanced rutin's water solubility and dissolution rate. In vitro tests demonstrated the complex's effective antibacterial and antibiofilm activities against Staphylococcus aureus and Pseudomonas aeruginosa strains, including resistant strains.

The introduction provides a comprehensive overview of the topic, particularly focusing on the challenges of treating ocular infections due to bacterial resistance and the potential of flavonoids like rutin in addressing these issues. However, there are additional areas that could be included to enhance the quality of the introduction

· Including more detailed statistics or data on the prevalence and impact of ocular infections globally could provide context on the scale of the problem.

· Including a brief overview of the current standard treatments for ocular infections, their limitations, and side effects would contrast well with the proposed solution.

Comments on the Quality of English Language

The manuscript is generally well-written with only a few minor areas for improvement:

Line 16 - "therapeutical" should be "therapeutic".

Line 33 - "If untreated properly" could be better phrased as "If not treated properly".

I would also ask the authors to pay particular attention to line breaks and incorrect use of hyphens throughout the manuscripts which were particularly evident throughout the abstract.

Author Response

Responses to Reviewer # 3

The manuscript outlines a study on a new therapeutic formulation for ocular infections, using a rutin/sulfobutylether-β-cyclodextrin (RTN/SBE-β-CD) inclusion complex. The major aims were to improve the solubility and therapeutic efficacy of rutin, a natural flavonoid with potential antibacterial properties. The study found that the RTN/SBE-β-CD complex significantly enhanced rutin's water solubility and dissolution rate. In vitro tests demonstrated the complex's effective antibacterial and antibiofilm activities against Staphylococcus aureus and Pseudomonas aeruginosa strains, including resistant strains.

The introduction provides a comprehensive overview of the topic, particularly focusing on the challenges of treating ocular infections due to bacterial resistance and the potential of flavonoids like rutin in addressing these issues. However, there are additional areas that could be included to enhance the quality of the introduction.

The authors are grateful to the Reviewer for his/her suggestions to improve the quality of the work. Below we reported the responses to the Reviewer’s comments. All changes made in the revised version of the manuscript are highlighted in yellow.

  • Including more detailed statistics or data on the prevalence and impact of ocular infections globally could provide context on the scale of the problem.
  • Including a brief overview of the current standard treatments for ocular infections, their limitations, and side effects would contrast well with the proposed solution.

As requested by the Reviewer, such information have been added in the text of the revised manuscript (see lines 33-77).

The manuscript is generally well-written with only a few minor areas for improvement:

Line 16 - "therapeutical" should be "therapeutic".

The word "therapeutical" has been changed in "therapeutic".

Line 33 - "If untreated properly" could be better phrased as "If not treated properly".

As requested by the Reviewer, the words "If untreated properly" have been changed in "If not treated properly" (line 33 of the revised manuscript).

I would also ask the authors to pay particular attention to line breaks and incorrect use of hyphens throughout the manuscripts which were particularly evident throughout the abstract.

Unfortunately, we cannot correct hyphens because they are generated by the MDPI template

Reviewer 4 Report

Comments and Suggestions for Authors

1.       Introduction should be reorganized. The most important aspects related to this topic should be clearly presented in order to provide a properly description of the state of art in this field. The utility of this study should be clearly highlighted in the manuscript. The introduction on Ocular infections and Flavonoids could be drastically reduced. The benefits, uses and limitations of rutin should be described. Research on the use of cyclodextrins to improve the physicochemical properties of insoluble substances should be introduced. some very recent studies below could be used as examples. https://doi.org/10.1016/j.jclepro.2023.137499, https://doi.org/10.1016/j.indcrop.2023.117974

https://doi.org/10.1016/j.foodchem.2020.127980

2.       Why SBE-β-CD was used? The author should explain the reasons for choosing SBE-β-CD.

3.       The preparation method of the physical mixture should be described.

4.       It was recommended to use different colored lines in Figure 3.

5.       It was suggested that TGA curve should be expressed in the form of weight change with temperature.

6.       Characteristic peaks should be labeled in Figure 7.

7.       Line 405: what did “the chemical shift of free and complexed RTN in a 1:1 molar ratio with SBE-β-CD.” mean?

Comments on the Quality of English Language

none

Author Response

Responses to Reviewer # 4

The authors are grateful to the Reviewer for his/her suggestions to improve the quality of the work. Below there are the responses to the Reviewer’s comments. All changes made in the revised version of the manuscript are highlighted in yellow.

Introduction should be reorganized. The most important aspects related to this topic should be clearly presented in order to provide a properly description of the state of art in this field. The utility of this study should be clearly highlighted in the manuscript. The introduction on Ocular infections and Flavonoids could be drastically reduced. The benefits, uses and limitations of rutin should be described. Research on the use of cyclodextrins to improve the physicochemical properties of insoluble substances should be introduced. some very recent studies below could be used as examples. https://doi.org/10.1016/j.jclepro.2023.137499, https://doi.org/10.1016/j.indcrop.2023.117974

https://doi.org/10.1016/j.foodchem.2020.127980

Introduction has been extensively rewritten, including more data on the prevalence and impact of ocular infections and a brief overview of the current standard treatments for ocular infections, their limitations, and side effects (see lines 33-67). Furthermore, the state of art concerning RTN/CD inclusion complexes has been treated (see lines 85-105 and references n. 30-42). The utility of our study has been highlighted in the revised version of the manuscript (see lines 106-118).

The introduction on flavonoids has been erased. Concerning ocular infection, we have been partially reduced the text. We were unable to reduce it further following requests from other reviewers.

In all cases, a substantial modification of the introduction has been made.

The benefits, uses and limitation of rutin have been described (see lines 70-81 of the revised manuscript).

We have inserted in the revised version of the manuscript the new references suggested by the Reviewer (see references n. 14-18)

  1. Why SBE-β-CD was used? The author should explain the reasons for choosing SBE-β-CD.

SBE-β-CD was selected for its higher water solubility and complexing ability with respect to native β-CD or other modified CDs usually present in pharmaceutical products. It shows high biocompatibility, non-toxicity and it is approved by FDA for parenteral administration. Fifteen products are on the market today containing SBE-β-CD.  Based on excellent properties showed by this macrocycle, a greater improvement of physical-chemical and biological properties of RTN could be expected using SBE-β-CD rather than other CDs. Furthermore, a very substantial literature is present concerning the complexation of RTN with native β-CD and different modified β-CD, but, to the best of our knowledge, very few papers describe the physical-chemical properties of RTN/SBE-β-CD inclusion complex and none of these reported its antimicrobial and antibiofilm activity neither in vitro nor in vivo.

We have added this explanation in the text (see lines 106-118 of the revised manuscript).

3. The preparation method of the physical mixture should be described.

The authors are grateful to the Reviewer for this observation and have added the preparation of the physical mixture in the paragraph 2.2. Preparation of RTN/SBE-β-CD inclusion complex and physical mixture.

  1. It was recommended to use different colored lines in Figure 3.

According to the suggestion of the Reviewer, the Figure 3 of the old manuscript has been corrected using different coloured lines (see Figure 2 of the revised manuscript).

  1. It was suggested that TGA curve should be expressed in the form of weight change with temperature.

As suggested by the Reviewer, the figure representing TGA curves has been corrected (see Figure 7 of the revised manuscript)

  1. Characteristic peaks should be labeled in Figure 7.

As suggested by the Reviewer, the characteristic peaks have been labelled (see Figure 8 of the revised manuscript)

  1. Line 405: what did “the chemical shift of free and complexed RTN in a 1:1 molar ratio with SBE-β-CD.” mean?

As requested by the Reviewer, the authors changed the sentence to better explain the meaning (see lines 378,379 in the revised version of the manuscript).

Round 2

Reviewer 2 Report

Comments and Suggestions for Authors

Accept in present form